# New Insights on the Role of Anti-PD-L1 and Anti-CTLA-4 mAbs on Different Lymphocytes Subpopulations in TNBC

**DOI:** 10.3390/cancers14215289

**Published:** 2022-10-27

**Authors:** Rosa Rapuano Lembo, Lorenzo Manna, Guendalina Froechlich, Emanuele Sasso, Margherita Passariello, Claudia De Lorenzo

**Affiliations:** 1Ceinge—Biotecnologie Avanzate s.c.a.r.l., Via Gaetano Salvatore 486, 80145 Naples, Italy; 2European School of Molecular Medicine, University of Milan, 20122 Milan, Italy; 3Department of Molecular Medicine and Medical Biotechnology, University of Naples “Federico II”, Via Pansini 5, 80131 Naples, Italy

**Keywords:** immunomodulatory mAbs, NK cells, TNBC, PD-L1, CTLA-4, combinatorial treatments

## Abstract

**Simple Summary:**

T cells have been considered, for a long time, key players in anti-cancer responses triggered by treatment with immune checkpoint inhibitors, but our recent studies also revealed the critical role of NK cells, due to the expression of ICs, such as PD-L1 and CTLA-4, on these immune cells also. Here, we investigated whether anti-PD-L1 or anti-CTLA-4 antibodies could modulate the effector functions of NK and T cell subpopulations differently in co-cultures with triple negative breast cancer cells. We found that the novel immunomodulatory antibodies, previously generated in our laboratory, more efficiently activate NK cells than the antibodies in clinical use, such as atezolizumab and ipilimumab. These results indicate that antibodies targeting different epitopes can have differential effects on different lymphocytes subpopulations and that novel combinations of mAbs could be suitable for therapeutic approaches aimed at activating not only T cells but also NK cells, especially for tumors lacking MHC.

**Abstract:**

Antibody-based cancer immunotherapy includes monoclonals against immune checkpoints (ICs), to modulate specific T cell responses against cancer. NK cells are a newly emerging target for immune checkpoint receptor inhibition in cancer immunotherapy, as ICs are also expressed on NK cells in various cancers. The latter cells are becoming attractive targets for cancer immunotherapy, as they are effector cells similar to CTLs, exerting natural cytotoxicity against primary tumor cells and metastasis, and they are able to distinguish tumor cells from healthy ones, leading to more specific anti-tumor cytotoxicity and reduced off-target effects. Thus, we decided to test the effects on isolated NK cells and T cell subpopulations of novel immunomodulatory mAbs, recently generated in our lab, in comparison with those in clinical use, such as ipilimumab and atezolizumab. Interestingly, we found that the novel anti-CTLA-4 (ID-1) and anti-PD-L1 (PD-L1_1) antibodies are able to induce NK cell activation and exert anti-tumor effects on TNBC cells co-cultured with NK cells more efficiently than the clinically validated ones, either when used as single agents or in combinatorial treatments. On the other hand, ipilimumab was found to be more effective in activating T cells with respect to ID-1. These findings indicate that antibodies targeting different epitopes can have differential effects on different lymphocytes subpopulations and that novel combinations of mAbs could be suitable for therapeutic approaches aimed at activating not only T cells but also NK cells, especially for tumors lacking MHC.

## 1. Introduction

Cancer immunotherapy is a ground-breaking strategy for tumor treatment, being one of the most prominent research areas for the development of new drugs. Immunotherapy consists of many different approaches, such as cancer vaccines, adoptive T cell (ATC) therapy and blockade of immune checkpoints [1].

Notably, monoclonal antibodies targeting immune checkpoints are emerging as innovative tools against several malignancies, owing to their specificity and reduced side effects, when compared to chemotherapy or radiotherapy [2,3]. Among these novel therapeutic agents, monoclonal antibodies against CTLA-4 (cytotoxic lymphocyte antigen-4), such as the clinically validated ipilimumab, or against PD-L1 (programmed cell death protein ligand 1), such as atezolizumab, have been approved by FDA for different types of cancer as first-line treatments, monotherapies or in combination with chemotherapy, showing great results in terms of both overall survival (OS) and progression-free survival (PFS) [4,5,6,7,8,9,10].

The mechanism of action of these immune checkpoint inhibitors is well understood: the inhibitory role of CTLA-4 and PD-L1 on the immune system, for instance, is exploited by the malignant cells to escape from the immune surveillance [11], hence, their blockade allows for a stronger response against the tumor. In particular, T cells play a pivotal role in this response, and both CD8+ cytotoxic lymphocytes and CD4+ helper T cells activity increases following the treatment with anti-CTLA-4 and anti-PD-L1 monoclonal antibodies [4,5]. Furthermore, despite being part of the innate immunity, recent studies highlighted that NK cells also have a major role in the outcome of immunotherapy-based treatments. Interestingly, NK cells express some immune checkpoints as well [12,13,14], and previous studies have shown how they could benefit from the treatment with anti-CTLA-4 or anti-PD-L1 monoclonal antibodies [13,14,15], even though a clear role has not yet been reported in the regulation of NK cells activity. Thus, this lymphocyte population could be an interesting target for understanding additional roles of ICs and developing new drugs.

The efficacy of immunomodulatory antibodies might be compromised when used as single agents, owing to different immune escape mechanisms [2], such as lack of tumor immunogenicity, suppressive tumor microenvironment, compensatory upregulation of other ICs, immunoediting, T cells permanent exhaustion and others [16,17,18,19,20]. Consequently, a large number of clinical trials have begun over the recent years in order to adopt combinatorial therapies in a wide range of cancers, which have shown remarkable improvements in overall response and clinical benefits, when compared to monotherapies [3,21,22,23,24,25,26,27]. However, on the other side, an enhancement of immune-related adverse effects (irAEs) has also been reported in several cases of patients treated with combinations of immune checkpoint inhibitors [28,29]. Hence, the evaluation of toxicity is an extremely important parameter to be considered when combinatorial treatments of immunomodulatory drugs are designed in order to avoid excessive irAEs.

Over the last few years, breast cancer was identified as a valid candidate for the therapy with immunomodulatory mAbs. Breast cancer is the most common type of cancer in women, displaying a wide heterogeneity at a molecular level, thereby having different clinical outcomes [30]. Specifically, triple-negative breast cancer (TNBC) shows not only an extremely aggressive and invasive phenotype but also resistance to the most common therapies currently approved for non-TNBC breast cancer, being one of the most challenging tumors to treat [23,24,25,31].

We previously generated two novel anti-CTLA-4 and PD-L1 mAbs, named ID-1 and PD-L1_1, respectively [32,33,34,35], and we used them in this study to test their effects as single agents or in combination on NK cells- or Pan T cells-enriched populations, co-cultured with TNBC or non-TNBC breast cancer cell lines, in comparison with the mAbs currently approved for clinical use, such as ipilimumab and atezolizumab. Our goal was to shed light on the differential involvement of NK and T cells populations stimulated by these new mAbs in order to identify the most effective way to exploit their biological properties for the treatment of this malignancy.

## 2. Materials and Methods

### 2.1. Antibodies

The following antibodies were used: anti-CTLA-4 ipilimumab mAb (Yervoy, Bristol Myers Squibb, NY, USA), anti-PD-L1 Atezolizumab mAb (InvivoGen, San Diego, CA, USA); commercial human anti-PD-L1 (G&P Biosciences, Santa Clara, CA, USA) and anti-CTLA-4 (R&D Systems, Minneapolis, MN, USA) Abs; anti-human IgG (H+L) HRP-conjugated Ab (Promega, Madison, WI, USA); anti-human IgG (Fab’)2 goat HRP-conjugated monoclonal antibody (Abcam, Biomedical Campus, Cambridge, UK).

ID-1 (anti-CTLA-4) and PD-L1_1 (anti-PD-L1) mAbs were produced and purified, as previously described [32,33,34,35], taking advantage of the enhanced cell line HEK293_ES1 expressing a long non-coding SINEUP RNA [36,37].

### 2.2. Cell Cultures

MDA-MB-231 breast cancer cells were cultured in Dulbecco’s Modified Eagle’s Medium (DMEM, Gibco, Life Technologies, Paisley, SCT, UK). BT-549 and BT-474 breast cancer cell lines were cultured in Roswell Park Memorial Institute 1640 Medium (RPMI 1640, Gibco, Life Technologies, Paisley, UK). MCF-7 cells were cultured in Modified Eagle’s Medium (MEM, Gibco, Life Technologies, Grand Island, NE, USA). Cell lines were purchased from the American Type Culture Collection (ATCC) and cultured in humidified atmosphere containing 5% CO_2_ at 37 °C. The media were supplemented with 10% (vol/vol) heat-inactivated fetal bovine serum (Sigma-Aldrich, St. Louis, MO, USA) and were used after addition of 50 U/mL penicillin, 50 μg/mL streptomycin, 2 nM L-glutammine (all from Gibco, Life Technologies, Paisley, UK). The cells were cultured as previously described [38].

### 2.3. Isolation of Human Peripheral Blood Mononuclear Cells

Human PBMCs, isolated and frozen as previously reported [32,39], were thawed out and used after resting overnight at 37° C in R10 medium consisting of RPMI 1640 supplemented with 10% inactivated FBS, 1% L-glutamine, 50 U/mL penicillin, 50 μg/mL streptomycin and 1% HEPES (Gibco, Thermo Fisher Scientific Waltham, MA, USA). After an overnight resting, the hPBMCs were collected in phosphate-buffered saline (PBS), counted by using the Muse Cell Analyzer (Merck Millipore, Darmdstadt, Germany) and resuspended for usage.

### 2.4. Isolation of NK and Pan T Cells

Human NK and Pan T cells isolation kits (MACS, Miltenyi Biotec, Bergisch Gladbach, Germany) were used to isolate the above-mentioned populations from unfractionated hPBMCs, according to the manufacturer’s guidelines. The cells were incubated with NK Cell Biotin-Antibody Cocktail or Pan T Cell Biotin-Antibody Cocktail for 5 min at 4 °C, and then they were incubated with NK cell or Pan T cell microbead cocktail for 10 min at 4 °C. NK or Pan T cells were separated by following the protocols, as previously described [34], and then the pellets were resuspended in R10 medium and counted by using a Muse cell analyzer.

### 2.5. Enzyme-Linked Immunosorbent Assays (ELISA)

To check the expression level of PD-L1 and CTLA-4 on cancer cells, cell ELISA assays were performed. Briefly, cells were plated in triplicates into a Nunc round-bottom 96-well plate at the density of 2 × 10^5^ cells/well and incubated with a blocking solution (PBS/BSA 6%) for 20 min at room temperature (RT). Then, cells were incubated in the absence or presence of commercial anti-PD-L1 (G&P Biosciences, Santa Clara, CA, USA) or anti-CTLA-4 (R&D Systems, Minneapolis, MN, USA) Ab at a concentration of 200 nM in PBS/BSA 3% buffer solution for 2 h at RT with gentle agitation. After the incubation with the primary antibodies, the plates were washed with PBS 1X and incubated with an appropriate HRP-conjugated secondary antibody for 1 h at RT. After extensive washes, 3,3′,5,5′-Tetramethylbenzidine (TMB) (Sigma-Aldrich, St. Louise, MO, USA) reagent was added for 10 min before quenching with an equal volume of 1 N HCl. Absorbance at 450 nm was measured by the Envision plate reader (Perkin Elmer, 2102, San Diego, CA, USA).

To check the expression level of PD-L1 and CTLA-4 on hPBMCs, NK and Pan T cells, cell ELISA assays were performed. Untreated cells and cells, previously activated with Staphylococcal Enterotoxin B (SEB, Sigma-Aldrich, St. Louise, MO, USA) at the concentration of 50 ng/mL for 4 different time intervals (24, 48, 72 and 96 h), were plated in triplicates into a Nunc round-bottom 96-well plate in suspension at the density of 2 × 10^5^ cells/well and incubated with a blocking solution (PBS/BSA 6%) for 20 min at RT. Since the cells were not adherent, they did not need to be fixed before blocking and they were recovered at each step by centrifugation at 1200 rpm for 10 min at RT. Then, cells were incubated in the absence or in the presence of commercial anti-PD-L1 or anti-CTLA-4 Abs at a concentration of 200 nM and treated as described above.

### 2.6. Cytotoxicity Assays and LDH Detection

Tumor cells co-cultured with hPBMCs, NK cells or Pan T cells were treated with the immunomodulatory mAbs used as single agents or in combination. Tumor cells were plated in 96-well flat-bottom plates at the density of 1 × 10^4^ cells/well, for 24 h at 37 °C. The novel ID-1 or PD-L1_1 mAbs, in parallel with the clinically validated ipilimumab and atezolizumab mAbs, were added to the above-mentioned co-cultures alone or in combination at the concentration of 100 nM. As negative controls, the co-cultured cells were analyzed in the absence of treatments or in the presence of a human unrelated IgG. After 48 h, the level of lactate dehydrogenase (LDH) released in the supernatants was evaluated as a marker of tumor cells lysis, as previously described [40], by using LDH detection kit (Thermofisher Scientific, Rockford, IL, USA).

### 2.7. Cytokine Secretion Assays

The levels of IL-2, IFNγ or Granzyme B, released in the supernatants of untreated or treated tumor cells co-cultured with NK or Pan T cells, were measured by ELISA assays. The kits provided by DuoSet ELISA (R&D Systems, Minneapolis, MN, USA) were used for the analyses, according to the producer’s recommendations.

### 2.8. Western Blotting Analyses

The hPBMCs and NK-enriched or Pan T-enriched populations were plated in 12-well plates at a density of 2 × 10^6^ cells/well and incubated for 48 h at 37 °C with SEB at a concentration of 50 ng/mL. Cells were collected by centrifugation at 1200 rpm for 10 min. In parallel, untreated BT-474 and MCF-7 tumor cells were detached from the plates and collected by centrifugation at 1200 rpm for 10 min. Cell pellets were resuspended in a lysis buffer containing 10 mM Tris-HCl (pH 7.4), 0.5% Nonidet-P-40, 150 mM NaCl, 1 mM Sodium orthovanadate (Sigma-Aldrich, St. Louise, MO, USA) and protease inhibitors (Roche, Indianapolis, IN, USA). After incubating on ice for 20 min, the extracts were clarified by centrifugation at 13,000 rpm for 20 min at 4 °C. Protein concentration was determined by performing a Bradford colorimetric assay (Bio-Rad, Hercules, CA, USA) before loading. Western blotting analyses were performed by incubating the membranes with the commercial human anti-PD-L1 (G&P Biosciences, Santa Clara, CA, USA) or commercial human CTLA-4 (R&D Systems, Minneapolis, MN, USA) primary antibodies, followed by the HRP-conjugated secondary antibodies. In order to normalize the signal intensities, the membranes were incubated with anti-vinculin mAb (Santa Cruz Biotechnology, Dallas, TX, USA) followed by goat anti-mouse polyclonal IgG HRP-conjugated secondary antibody (Sigma-Aldrich, Burlington, MA, USA).

### 2.9. Statistical Analyses

The standard deviation (±SD) was determined by using an automatic graphing calculator on the basis of the results obtained by at least three independent experiments. The SD was calculated and added as vertical lines with a little cross bar at the top and bottom. Statistical significance was determined by Student’s *t*-test (two variables) and the *p* value was established as *** *p* 0.001; ** *p* < 0.01; * *p* < 0.05.

## 3. Results

### 3.1. Analysis of PD-L1 and CTLA-4 Expression on Tumor Cells and Immune Cell Subpopulations

Since our previous data have shown that CTLA-4 and PD-L1 ICs are expressed at satisfactory levels on MDA-MB-231 and BT-549 TNBC cells [40], we investigated whether these targets were also expressed on a malignant non-TNBC subtype of breast cancer, such as the BT-474 cell line, which could represent a good model for highlighting differences among different breast cancer subtypes for the emerging anti-cancer drugs [41]. To this aim, cell ELISA assays were performed to evaluate the levels of ICs on cell surface by incubating the cells with a commercial anti-PD-L1 or anti-CTLA-4 mAb, used at saturating concentrations. As a negative control, MCF-7 cells were used in parallel assays, as they were previously found to express very low levels of PD-L1 and CTLA-4 [42]. As shown in Figure 1A, BT-474 cells express levels of PD-L1 and CTLA-4 comparable to those observed on BT-549 cells, thus, indicating that there are no significant differences in the expression of ICs on TNBC, compared to non-TNBC cells. The higher expression of PD-L1 and CTLA-4 on BT-474, with respect to MCF-7 cells observed by Cell ELISA, was confirmed by Western blotting analyses of cell extracts (Appendix A).

Furthermore, we also checked the expression of these IC targets on different immune cells subpopulations and investigated whether they were constitutively expressed or modulated over time following stimulation. To this aim, cell ELISA assays were performed on total hPBMCs and on the two isolated subpopulations: NK cells-enriched population and Pan T cells-enriched population. The mentioned immune cells were stimulated with staphylococcal enterotoxin B (SEB) at increasing time intervals up to 96 h; then cell ELISA measured the levels of PD-L1 and CTLA-4 and their variation during the time-course. As a negative control, unstimulated lymphocytes were used in parallel assays (). As a further control, a Western blotting analysis of cell extracts of hPBMCs, NK and PanT cells stimulated with SEB for 48 h was performed in parallel (Appendix A). The results reported in Figure 1B and Appendix A, according to the literature [12,14,43,44,45,46], show that these two IC targets are expressed on the surface of stimulated NK and Pan T cells at comparable levels, with respect to total hPBMCs, whereas the expression levels of PD-L1 and CTLA-4, when the unfractionated immune cells were not stimulated, were much lower, as previously reported [32,34]. Interestingly, we found that the levels of CTLA-4 in these two populations were already significant at time 0 and only slightly increased over time after stimulation.

### 3.2. Effects of Novel Anti-PD-L1 and CTLA-4 mAbs on Cytokine Release by hPBMCs Subpopulations

Two novel human mAbs specifically targeting PD-L1 and CTLA-4, respectively, PD-L1_1 and ID-1, were recently generated in our laboratory and characterized by in vitro and in vivo studies [32,33,34]. Previous data showed that these novel mAbs are able to activate unfractionated hPBMCs, when stimulated with SEB, with more potent effects, compared to those of atezolizumab and ipilimumab mAbs, currently approved for clinical use, either when used as single agents or in combination [24,40]. Here, in order to clarify how each hPBMCs subpopulation responds to the effects of the novel mAbs, we repeated these experiments on NK-enriched or Pan T-enriched subpopulations, evaluating the release of some cytokines as specific markers of activation of each subpopulation: IFNγ for both subpopulations, Granzyme B for NK cell activation and IL-2 for Pan T cell response. Thus, each population, following stimulation with SEB, was treated with the novel PD-L1_1 or ID-1 mAbs or in parallel assays with the FDA-approved atezolizumab or ipilimumab, used alone or in combination for 66 h, as previously described for unfractionated hPBMCs [24]. This time interval was chosen as it corresponds to the peak of cytokines secretion for unfractionated hPBMCs [32]. Cells, untreated or treated with an unrelated mAb, were used as negative controls. The results, reported in Figure 2, show that the treatment of NK cells-enriched population with the novel mAbs either used alone or in combination determines a higher release of IFNγ and Granzyme B in the supernatants, with respect to that observed in the treatment with the clinically approved mAbs. Interestingly, on the other hand, the combination of atezolizumab plus ipilimumab leads to a more efficient activation of the Pan T cells-enriched population, with higher IFNγ and IL-2 release, with respect to that observed with the combination of novel PD-L1_1 and ID-1 mAbs.

Thus, we showed a differential activity of the indicated anti-PD-L1 and anti-CTLA-4 mAbs on the distinct lymphocytes subpopulations, which seems to differentially modulate the pathways involved in their activation. We cannot exclude that some LDH release could also be mediated by NK T cells that could be present in both the cell subpopulations, but the percentage of these cells is very low even though they express CTLA-4 and PD-L1, as previously described [47,48]; therefore, they could respond to these antibodies.

### 3.3. Effects of Anti-PD-L1 and Anti-CTLA-4 mAbs on non-TNBC Cells Co-Cultured with the Two Different hPBMCs Subpopulations

In our previous experiments, we proved that the novel generated anti-PD-L1 and anti-CTLA-4 mAbs had a cytotoxic activity stronger than FDA-approved ipilimumab and atezolizumab [24,40] when used on TNBC cells co-cultured with hPBMCs. Starting from these results, we also decided to investigate their cytotoxic effects on the non-TNBC BT-474 cell line when co-cultured with total hPBMCs or its derived NK or Pan T cells-enriched subpopulations. Therefore, BT-474 were co-cultured with unfractionated hPBMCs, isolated NK cells or Pan T cells subpopulations at immune effector cells: tumor target cells ratio of 3:1 and treated with PD-L1_1, ID-1, atezolizumab or ipilimumab mAb, used alone or in combination, for 48 h at 37 °C. As negative controls, TNBC cells co-cultured with hPBMCs, in the absence of treatment or in the presence an unrelated IgG mAb, were analyzed. After 48 h, the cell supernatants were collected and the LDH release was measured to evaluate the tumor cell lysis.

The results (Figure 3) show that, in the supernatants of BT-474 co-cultured with unfractionated hPBMCs and NK cells, the LDH release was higher when the cells were treated with the combination of the novel mAbs, with respect to the combination of the clinically validated mAbs, reaching about 60% of cell lysis, compared to 20–40% obtained with atezolizumab plus ipilimumab (Figure 3A,B), according to the data obtained from previous studies on TNBC cells [24]. Instead, when BT-474 were co-cultured with Pan T cells, we observed a higher release of LDH in the presence of atezolizumab plus ipilimumab treatment, reaching 70% of cell lysis, compared to 50% obtained with ID-1 plus PD-L1_1 (Figure 3C). As a negative control, BT-474 tumor cells were treated with the antibodies in the absence of immune cells and no significant tumor cells lysis was observed, thus, confirming that the immunomodulatory mAbs exert their effects by activating the immune cells.

The levels of IFNγ, IL-2 and Granzyme B released in the supernatants of co-cultures of BT-474 with hPBMCs, Pan T or NK cells were also measured, and the obtained data are reported in Figure 4. The release of IFNγ measured in the presence of total hPBMCs is comparable for the two combinatorial treatments. On the opposite, IFNγ and Granzyme B cytokines released from co-cultures with NK cells are higher when the cells are treated with the combination of novel PD-L1_1 with ID-1 mAbs (Figure 4B,D), whereas a higher release of IFNγ and IL-2 cytokines is observed in the co-cultures with Pan T cells when treated with the combination of atezolizumab with ipilimumab (Figure 4C,E). These results are in line with the different cytotoxic effects exerted by these two combinatorial treatments and reported in Figure 3.

### 3.4. Cytotoxic Effects of Immunomodulatory mAbs on TNBC Cells Co-Cultured with the Two Different hPBMCs Subpopulations

To further confirm whether the novel PD-L1_1 and ID-1 mAbs can differentially modulate the activation of the two considered lymphocyte subpopulations, we analyzed their effects also on TNBC cells, with respect to those observed with the clinically approved mAbs. To this aim, MDA-MB-231 and BT-549 TNBC cell lines were co-cultured with NK- or Pan T-enriched cell populations (effector: target cells ratio of 3:1) and treated with the novel mAbs or the clinically approved ones, used alone or in combination, for 48 h at 37 °C. Then, we collected the cell supernatants and measured the LDH release as a marker of cytotoxicity and calculated the % of cells lysis (Figure 5). According to the data described above, for both the TNBC cell lines, we observed a significant increase in the cell lysis in the co-cultures with NK cells when they were treated with the combination of PD-L1_1 and ID-1, with respect to that observed with atezolizumab and ipilimumab combination (see Figure 5A,B for MDA-MB-231, and Figure 5C,D for BT-549 cells). In particular, the lysis of MDA-MB-231 reached the value of 80% after combinatorial treatment with the novel mAbs (Figure 5A), compared with 46% of lysis reached with FDA-approved mAbs. On the other hand, supernatants of tumor cells co-cultured with Pan T showed a slightly higher release of LDH after treatment with the combination of the clinically validated mAbs (that reached 50% of lysis in the case of MDA-MB-231 cell line (Figure 5D)), with respect to about 34% observed after treatment with PD-L1_1 plus ID-1.

As negative controls, MDA-MB-231 and BT-549 TNBCs tumor cells were treated with the antibodies in the absence of immune cells and no significant tumor cell lysis was observed, thus, confirming that the immunomodulatory mAbs exert their effects by activating the immune cells.

In order to evaluate the variation in the activation of the two subpopulations when subjected to the action of the two different pairs of antibodies, we analyzed cytokines secretion by ELISA assays on the supernatants. Thus, the levels of IFNγ, IL-2 and Granzyme B released in the supernatants of MDA-MB-231 (Figure 6) and BT-549 (Figure 7) cells co-cultured with NK or Pan T cells were measured. As shown in Figure 6A and Figure 7A, IFNγ release from both TNBC cell lines co-cultured with NK cells was higher when PD-L1_1 or ID-1 and their combination were added, with respect to the clinically approved mAbs; similar results were observed for the secretion of Granzyme B in the supernatants of the co-cultures with MDA-MB-231 cells (Figure 6C).

On the contrary, a higher release of IFNγ and IL-2 was observed in the supernatants of the tumor cells co-cultured with Pan T when treated with atezolizumab plus ipilimumab, reaching a concentration of IFNγ of about 3000 pg/mL and 2600 pg/mL, respectively, in the co-cultures of MDA-MB-231 and BT-549 cell lines (about 2-fold higher, with respect to the combination of the novel mAbs) (Figure 6B and Figure 7B). Moreover, IL-2 achieved a level of 8000 pg/mL and 9570 pg/mL when MDA-MB-231 and BT-549 cells, respectively, were treated with the combination of the clinically validated mAbs, whereas a slightly lower concentration of 5600 pg/mL and 6370 pg/mL was reached when the combination of PD-L1_1 and ID-1 was used on the same cell lines (Figure 6D and Figure 7D).

As a negative control (see Appendix A), LDH release and cytokines secretion, following treatment with the immunomodulatory mAbs, were also measured when unfractionated hPBMCs (Appendix A) or the two subpopulations (Appendix A) were co-cultured with the breast cancer MCF-7 cell line, which is PD-L1-negative and expresses low levels of CTLA-4 [42]. No significant differential effects were observed in the treatment of MCF-7 cells with the different antibodies or their combinations in the presence of both the two NK and T cell subpopulations.

## 4. Discussion

Monoclonal antibody-based immunotherapy is an innovative approach able to restore the immune system capability of recognizing and eliminating tumor cells in order to treat several types of malignancies that are resistant to conventional treatments [1,2,3]. Different immune cells subpopulations are involved in the response triggered by these therapeutic agents, mainly represented by T reg, CD8 and CD4 T cells [4,5], but recent studies have shown that also NK cells actively participate in the successful outcome of anti-cancer treatment [12,13,14,15].

Each monoclonal antibody could have a distinct mechanism of action, depending on several factors, such as the class of the immunoglobulin, the recognized epitope, or the antibody affinity for its target, thus, they could differently affect diverse immune cell populations. Therefore, the knowledge of their effects on immune cells is crucial to predict the outcome of the therapy and to exploit the biological properties of these agents.

Recently, two novel anti-PD-L1 and anti-CTLA-4 mAbs, named PD-L1_1 and ID-1, respectively, were generated in our lab [32,33,34,35]. In this study, we investigated their ability to activate NK-enriched and Pan T-enriched subpopulations, isolated from unfractionated hPBMCs, and evaluated their cytotoxic effects on a panel of breast cancer cell lines, both TNBC and non-TNBC, co-cultured with each isolated subpopulation of immune cells. Their effects were compared to those observed with the clinically validated anti-PD-L1 and anti-CTLA-4 mAbs, atezolizumab and ipilimumab [4,5,6,7,8,9] (the latter endowed with the same isotype of ID-1 but targeting a distinct epitope [24]), respectively, to clarify whether they have similar or different activity on the different subpopulations.

Firstly, we measured the expression levels of PD-L1 and CTLA-4 on breast tumor cells and immune cells subpopulations following activation with SEB. We show here for the first time that NK cells express higher levels of CTLA-4 than Pan T cells, especially at 48 h of activation, whereas PD-L1 expression is similar over time between the two cell populations. Then, we evaluated the cytokines secretion of NK cells or Pan T cells following stimulation with SEB and treatment with the two novel antibodies, used alone or in combination, with respect to the effects observed after treatment with the clinically validated mAbs, atezolizumab and ipilimumab. The results showed that the treatment with the novel mAbs is able to activate the NK-enriched subpopulation more efficiently than the one with the antibodies approved for clinical use; in particular, it determines a higher release of IFNγ when the mAbs are used as single agents and higher levels of both IFNγ and Granzyme B when used in combination. On the other hand, the treatment with atezolizumab and ipilimumab, or their combination, led to a more marked activation of the Pan T cells subpopulation, with higher levels of both IFNγ and IL-2 released in the supernatants.

To test whether the differential effects on immune cells cause different anti-tumor effects, we investigated on the cytotoxic activity of these mAbs in co-cultures of each immune cell population with three different breast cancer cell lines, triple negative (TNBC) MDA-MB-231 and BT-549 cell lines and BT-474 (non-TNBC) cell line. The LDH release was measured after 48 h of treatment to evaluate tumor cell lysis, and the cytokines secretion in the supernatants was evaluated as markers of lymphocytes activation. The data showed that the treatment of NK-enriched population with PD-L1_1 and ID-1 combination determined a stronger LDH release for all the tested tumor cell lines, when compared to that observed for the treatment with atezolizumab plus ipilimumab, along with higher secretion of IFNγ and Granzyme B, especially for the BT-474 and the MDA-MB-231 cell lines. On the contrary, co-cultures with the Pan T-enriched population led to a more marked LDH, IFNγ and IL-2 release after treatment with the combination of the clinically validated atezolizumab and ipilimumab.

To explain why the combination of the novel anti-PD-L1 and anti-CTLA-4 mAbs, PD-L1_1 and ID-1, showed the ability to activate more efficiently the NK-enriched population, with respect to atezolizumab plus ipilimumab, several factors might be considered, including the different epitopes recognized by these mAbs [24].

Indeed, recent studies showed that NK cells benefit from the treatment with some anti-PD-L1 mAbs, owing to a molecular pathway, mediated by p38 that leads to a stronger activation and prevents the exhaustion of these cells [43], which could be differentially affected by different mAbs. We exclude that the differential properties of the two anti-PD-L1 mAbs can also be related to their structures, that are identical, or to their different isotypes, as PD-L1_1 is an IgG4 devoid of effector functions, in comparison with the IgG1 format of atezolizumab, which could be instead recognized by the CD16 receptor of NK cells. Despite this, it seems that PD-L1_1 is more active than atezolizumab on NK cells.

Concerning the novel anti-CTLA-4 mAb, ID-1, previous data displayed different features when compared to the FDA-approved ipilimumab, owing to its different epitope and reduced antagonistic activity on T cells, even though it was found still able to deplete the inhibitory T-reg cells population by activating NK cells, as it is an IgG1, like ipilimumab, and thus it should have a similar effect on CD16 [34].

These new findings could improve the therapeutic approaches by using novel combinatorial treatments involving antibodies that can modulate both the activation of immune T cells and NK cells in order to increase their potency and lead to a stronger response against tumor cells. Indeed, the activation of tumor-infiltrating NK cells could be particularly beneficial for tumors characterized by the loss of MHC expression due either to tumor escape mechanisms or to previous ICIs or other anti-tumor therapies.

## 5. Conclusions

We investigated on the expression of the immune checkpoints PD-L1 and CTLA-4 on the surface of isolated immune cell subpopulations, and we found that NK cells constitutively express high levels of CTLA-4, whereas the PD-L1 levels are comparable between the two subpopulations. Thus, we tested the effects of ICIs targeting CTLA-4 and PD-L1 on these cells, and we found that the combination of novel PD-L1_1 and ID-1 mAbs, previously generated in our laboratory, leads to an enhanced activation and anti-tumor activity of NK cells in co-cultures with breast cancer cell lines, with respect to the clinically validated atezolizumab and ipilimumab that, on the other hand, activate Pan T cells more efficiently.

Therefore, these novel anti-PD-L1 and anti-CTLA-4 mAbs could be useful, especially against tumors with reduced expression of MHC complex, where NK cells could serve as the main players in anti-tumor response, since T cells activity might be impaired.

## 6. Patents

One antibody, described in this study, was included in the patent application PCT/EP20 19/057239.

## Figures and Tables

**Figure 1 cancers-14-05289-f001:**
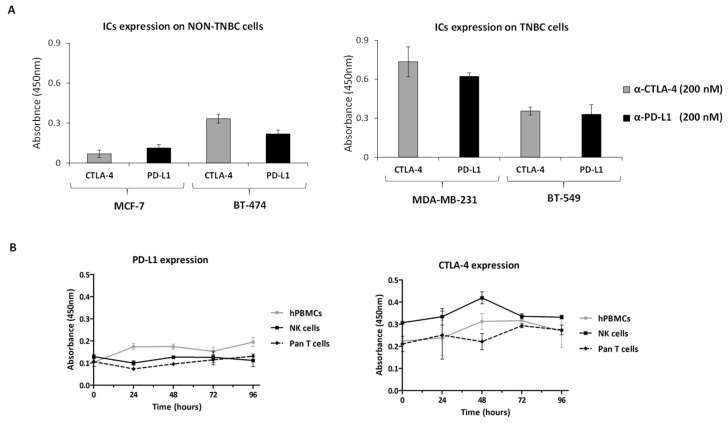
Expression of PD-L1 and CTLA-4 on breast cancer and immune cells. (**A**) Cell ELISA assays were performed by incubating BT-474, MCF-7 ((**A**), panel on the left), MDA-MB-231 or BT-549 ((**A**), panel on the right) cells with the commercial anti-PD-L1 or anti-CTLA-4 mAb at the indicated concentrations. (**B**) hPBMCs (grey curves), NK cells-enriched (black curves) or Pan T cells-enriched populations (dashed black curves) were incubated with SEB (50 ng/mL) for different time intervals (24, 48, 72 and 96 h) and then tested with the anti-PD-L1 or anti-CTLA-4 commercial mAb (200 nM) to measure the cell surface expression of the indicated ICs. The values were reported as the mean of at least three determinations obtained in three independent experiments. Error bars depicted means ± SD.

**Figure 2 cancers-14-05289-f002:**
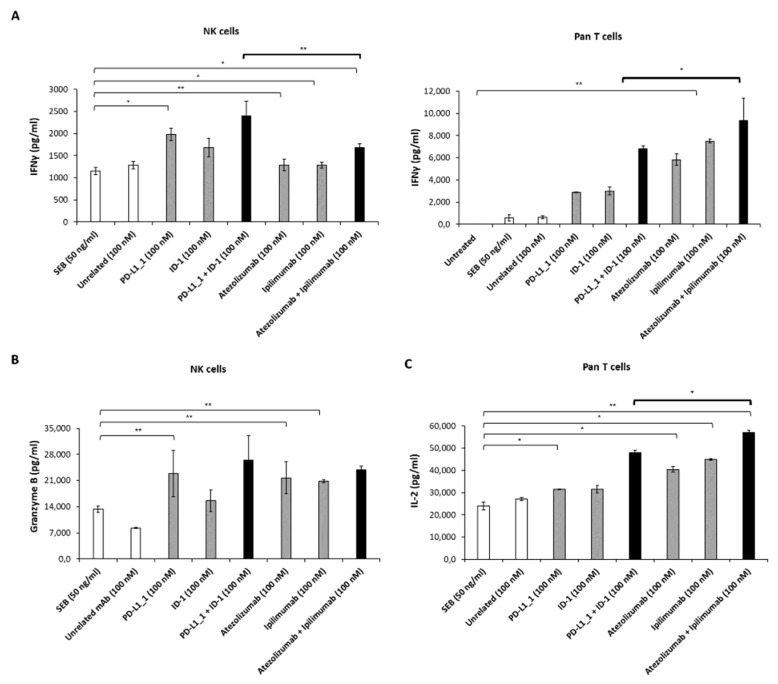
Effects of novel mAbs on the activation of stimulated lymphocytes. NK- and Pan T-enriched populations were treated with SEB for 66 h, in the absence (white bars) or in the presence of the anti-PD-L1 (PD-L1_1 or atezolizumab) or the anti-CTLA-4 (ID-1 or ipilimumab) mAbs, used as single agents (light grey bars) or in combination (black bars). The levels of IFNγ (**A**), Granzyme B (**B**) and IL-2 (**C**) were measured by ELISA assays (performed as described in Materials and Methods). Cells untreated or treated with an unrelated mAb were used as negative controls. Values were reported as the mean of at least three determinations obtained in three independent experiments. Error bars depict means ± SD. P-values were calculated by comparing each treatment with untreated co-cultured cells or comparing the two combinations with each other and reported as ** *p* < 0.01; * *p* < 0.05.

**Figure 3 cancers-14-05289-f003:**
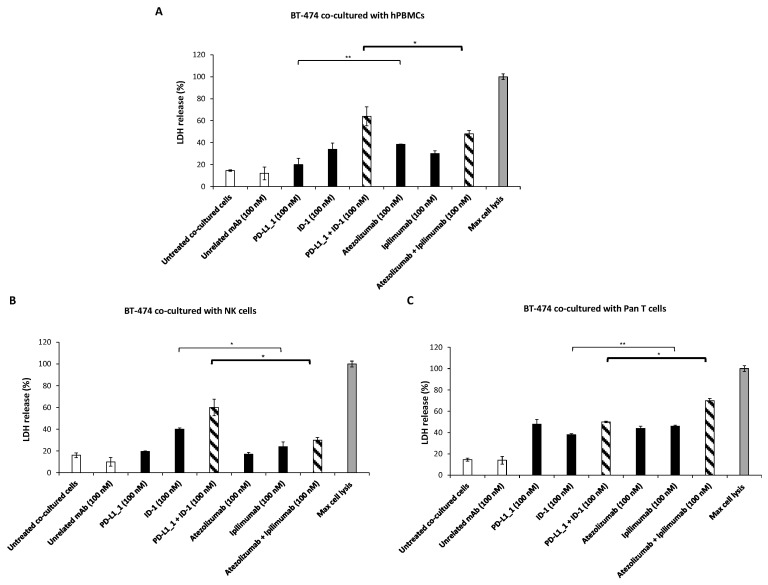
Cytotoxic effects of immunomodulatory mAbs on non-TNBC cells co-cultured with total hPBMCs or its derived NK and T cells subpopulations. Cells were cultured with the indicated immune cells in the absence or presence of the novel immunomodulatory mAbs, used alone (black bars) or in combination (striped bars) at the indicated concentrations for 48 h. The clinically validated atezolizumab and ipilimumab were used for comparison in parallel assays. Untreated cells or cells treated with an unrelated mAb were used as negative controls (white bars). After treatment, the release of LDH was measured by collecting the supernatants from the tumor cells co-cultured with total hPBMCs (**A**), NK (**B**) or Pan T-enriched populations (**C**). The tumor cell lysis was expressed as %, with respect to the 100% of lysis (grey bars) obtained by incubating the cells with 10% Triton X100. Cell lysis values were reported as the mean of at least three determinations obtained in three independent experiments. Error bars depict means ± SD. P-values were calculated by comparing each treatment with untreated co-cultured cells or comparing the two combinations with each other and reported as ** *p* < 0.01; * *p* < 0.05.

**Figure 4 cancers-14-05289-f004:**
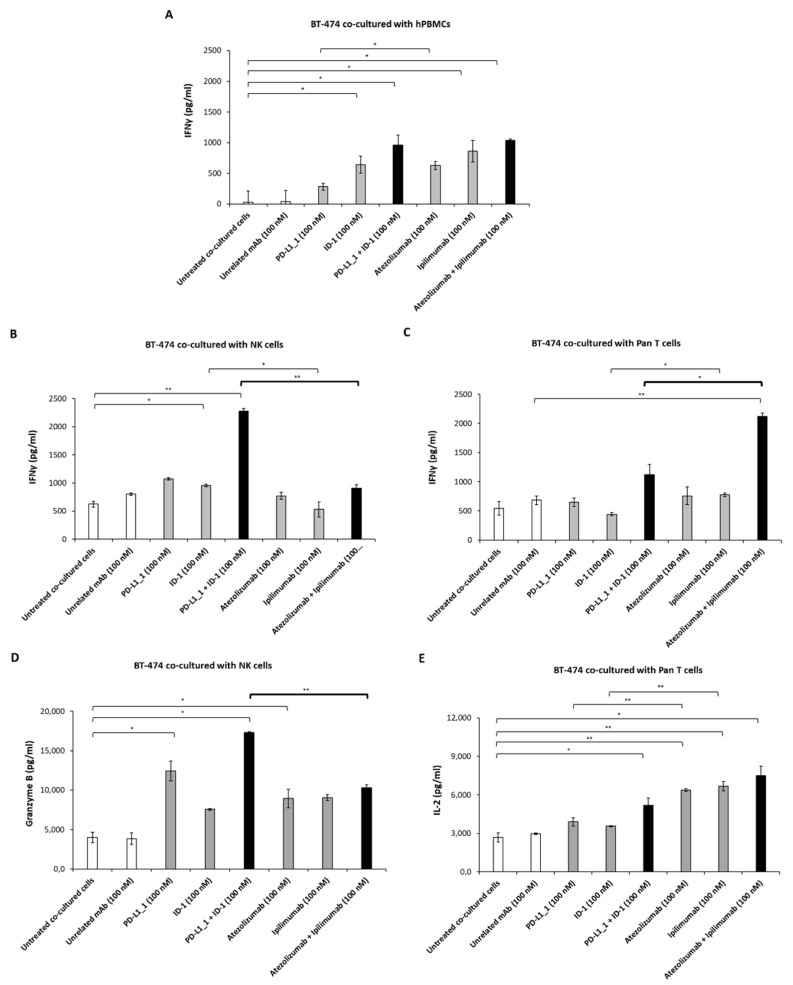
Effects of anti-PD-L1 and anti-CTLA-4 mAbs on cytokines secretion from co-cultures of non-TNBC cells with hPBMCs, NK or Pan T cells. BT-474 cells were co-cultured with total hPBMCs (**A**), NK-enriched (**B**,**D**) or Pan T-enriched (**C**,**E**) cell populations in the absence or in the presence of the anti-CTLA-4 and anti-PD-L1 mAbs, used alone (grey bars) or in combination (black bars), for 48 h at 37 °C. Untreated cells or cells treated with an unrelated IgG were used as negative controls (white bars). The release of IFNγ, IL-2 and Granzyme B were measured in the supernatants of the cells, treated as indicated and performed by following the manufacturer’s recommendations (DuoSet ELISA kit, ThermoFisher Scientific). Cytokine concentration values were reported as the mean of at least three determinations obtained in three independent experiments. Error bars depict means ± SD. P-values were calculated by comparing each treatment with untreated co-cultured cells or comparing the two combinations with each other and reported as ** *p* < 0.01; * *p* < 0.05.

**Figure 5 cancers-14-05289-f005:**
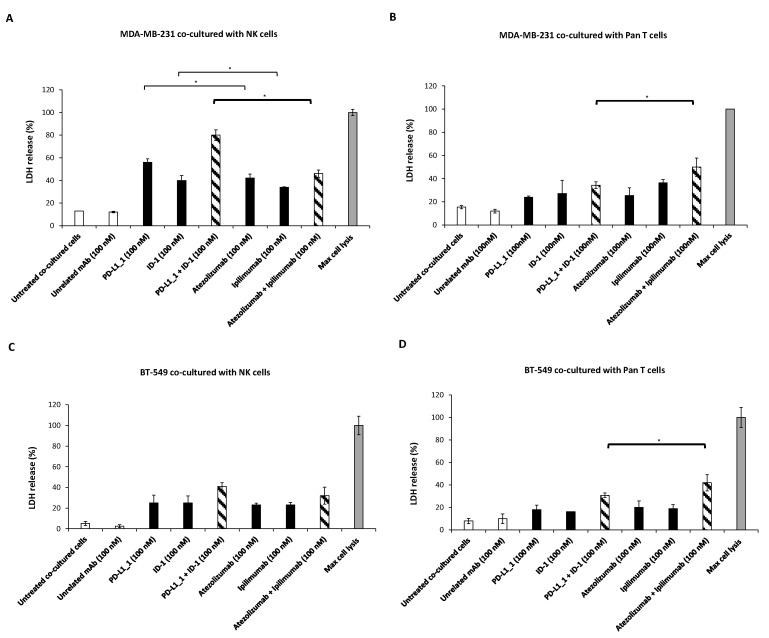
Cytotoxic activity and cytokine secretion induced by the novel immunomodulatory mAbs on MDA-MB-231 and BT-459 TNBC cells co-cultured with NK or Pan T-enriched subpopulations. MDA-MB-231 cells were co-cultured with NK (**A**) or Pan T (**B**); BT-549 cells were co-cultured with NK (**C**) or Pan T (**D**), in the absence or presence of PD-L1_1 or ID-1 mAb, used alone (black bars) or in combination (striped bars), at the indicated concentrations for 48 h. Atezolizumab and ipilimumab were used for comparison in parallel assays. Untreated cells or cells treated with an unrelated mAb were used as negative controls (white and grey bars). The release of LDH was detected in the supernatant of the cells and expressed as previously described. The values were reported as the mean of at least three determinations obtained in three independent experiments. Error bars depict means ± SD. P-values were calculated by comparing each treatment with untreated co-cultured cells or comparing the two combinations with each other and reported as * *p* < 0.05.

**Figure 6 cancers-14-05289-f006:**
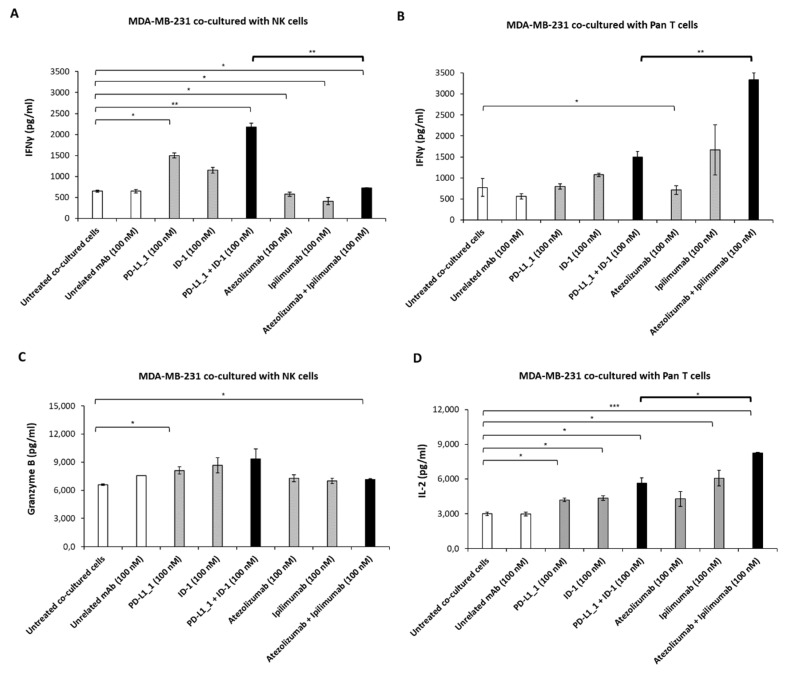
Cytokine secretion induced by the novel mAbs on co-cultures of MDA-MB-231 cells with NK or Pan T subpopulations. Tumor cells were co-cultured with NK (**A**) or Pan T cells (**B**) in the absence or in the presence of anti-CTLA-4, anti-PD-L1 mAbs used alone (grey bars) or in combination (black bars). Untreated hPBMCs or treated with an unrelated IgG were used as negative controls (white bars). The levels of IFNγ (**A**,**B**), Granzyme B (**C**) and IL-2 (**D**) secreted were measured by ELISA kits, as described above. The values were reported as the mean of at least three determinations obtained in three independent experiments. Error bars depict means ± SD. P-values were calculated by comparing each treatment with untreated co-cultured cells or comparing the two combinations with each other and reported as *** *p* < 0.001; ** *p* < 0.01; * *p* < 0.05.

**Figure 7 cancers-14-05289-f007:**
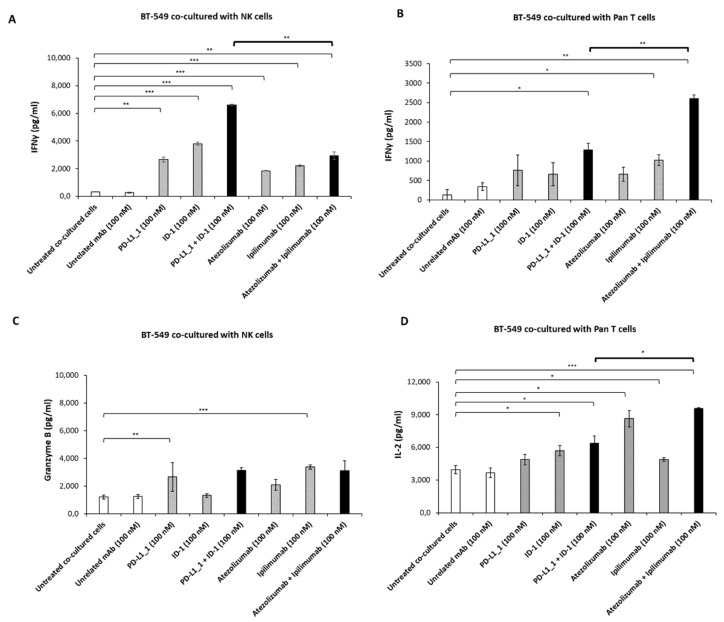
Cytokine secretion induced by immunomodulatory mAbs on co-cultures of BT-549 cells with NK or Pan T subpopulations. Tumor cells were co-cultured with NK (**A**) or Pan T cells (**B**) in the absence or in the presence of anti-CTLA-4, anti-PD-L1 mAbs used alone (grey bars) or in combination (black bars). Untreated hPBMCs or treated with an unrelated IgG were used as negative controls (white bars). The levels of IFNγ (**A**,**B**), Granzyme B (**C**) and IL-2 (**D**) secreted were measured by ELISA kits, as described above. The values were reported as the mean of at least three determinations obtained in three independent experiments. Error bars depict means ± SD. P-values were calculated by comparing each treatment with untreated co-cultured cells or comparing the two combinations with each other and reported as *** *p* < 0.001; ** *p* < 0.01; * *p* < 0.05.

## Data Availability

Not applicable.

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
