# Peer review of "New Insights on the Role of Anti-PD-L1 and Anti-CTLA-4 mAbs on Different Lymphocytes Subpopulations in TNBC"

_cancers, 2022, doi:10.3390/cancers14215289_

Round 1

Reviewer 1 Report

The article has been written in a consistent and compact way. It concerns the continuation of the authors’ work on development of novel molecules in anty-tumor immunotherapy. It gives promising results in researching on more specific anty-tumor drugs. The authors described laboratory results on novel molecules in comparison of already registered medicines. The work which has been done by the authors confirmed that there is no class effect of immune checkpoints inhibitors. I have to accent, many molecules tested in laboratories did not pass through clinical trials. There is strong need to test the described novel molecules in clinical trials. 

Author Response

We thank the reviewer for his/her kind words on our work and paper. We fully agree with his/her comments on the strong need of novel drugs for TNBC and we really hope that some of the reported molecules will be tested in clinical trials in the future.

Reviewer 2 Report

The manuscript extends previous publications by this group characterizing novel human CTLA4 and PD-L1 antibodies that differ from the currently used therapeutic antibodies in their ability to modulate signaling in T or NK cells rather than simply blocking binding of these immune checkpoints with their counter receptors. The major conclusion seems to be that the combined novel antibodies better stimulate killing of tumor cells by donor NK cells than by donor pan T cells. For reasons addressed below, the T cell data may have less clinical relevance than the NK cell data. In addition, any new antibody targeting these immune checkpoints would first be tested in patients as a single agent before entering a combined agent trial. Therefore, it is important to present statistics for the single agent efficacy data from the in vitro coculture experiments.

11.       The manuscript refers to earlier publications that included extensive descriptions of the novel antibodies, but it would be helpful to summarize for readers some details including whether all are IgG1, whether their Fc effector activities differ from those of the approved clinical antibodies, receptor dimerization activity as discussed in ref #34, direct cytotoxic effects on the breast cancer cells, and other relevant properties that differ.

22.       Fig 3 and 5 include statistics for the combined novel antibodies versus Atezolizumab and Ipilimumab but not for single comparisons of ID-1 versus Ipilumimab and PD-L1_1 versus Atezolizumab. Are the respective single antibody comparisons significant in each panel?

33.       Same statistics are needed for Fig 4.

44.       In Fig 3 and 5, it is reasonable that most of the tumor cell killing by hPBMCs would be mediated by NK cells rather than T cells because healthy donors should lack CD8 T cells with TCRs specific for relevant breast cancer antigens with or without correct MHC restriction. Could the observed LDH release be mediated by NKT cells in the pan-T preparation rather than CD8 T cells? If so, do NKT express CTLA4 and PD-L1 and respond to these antibodies?

55.       The relevant question for comparing T cell modulation by the new antibodies versus the existing therapeutic antibodies is: how well do they increase LDH release when pan T cells obtained from breast cancer patients are incubated with their own tumor cells?  This question could be better addressed using existing CD8 T cell lines with appropriate TCRs that recognize known breast cancer antigens.

66.       Given that Fig 1A,B documents some expression of CTLA-4 and PD-L1 on the breast cancer cells, and previous publications have reported direct cytotoxic activities of these antibodies for other cell types, do the antibodies cause any LDH release when incubated with these cell lines in the absence of PBMCs?

Author Response

We wish to thank you and the Reviewers for the useful comments and suggestions on our manuscript (ID: cancers-1957876) entitled "New Insights on the Role of Anti-PD-L1 and Anti-CTLA-4 mAbs on different Lymphocytes Subpopulations in TNBC ". We have fully revised the manuscript by inserting most of the requested modifications or experiments and by preparing new figures.

Please find below a point-by-point reply to the Reviewers comments (reported in italic). Major changes are highlighted by using the "Track Changes" function within the main body of the text.

  1. The manuscript refers to earlier publications that included extensive descriptions of the novel antibodies, but it would be helpful to summarize for readers some details including whether all are IgG1, whether their Fc effector activities differ from those of the approved clinical antibodies, receptor dimerization activity as discussed in ref #34, direct cytotoxic effects on the breast cancer cells, and other relevant properties that differ.

We thank the reviewer for his/her comments. We have added the requested details relative to the Fc and isotypes of the mAbs in a new paragraph inserted in the Discussion section (see p.15, lines 488-497).

  1. Fig 3 and 5 include statistics for the combined novel antibodies versus Atezolizumab and Ipilimumab but not for single comparisons of ID-1 versus Ipilumimab and PD-L1_1 versus Atezolizumab. Are the respective single antibody comparisons significant in each panel?

We thank the reviewer for his/her suggestions. We have included statistics for single comparisons in the new Figures 3 and 5.

  1. Same statistics are needed for Fig 4.

We thank the reviewer for his/her suggestions. We have included statistics for single comparisons in the new Figure 4.

  1. In Fig 3 and 5, it is reasonable that most of the tumor cell killing by hPBMCs would be mediated by NK cells rather than T cells because healthy donors should lack CD8 T cells with TCRs specific for relevant breast cancer antigens with or without correct MHC restriction. Could the observed LDH release be mediated by NKT cells in the pan-T preparation rather than CD8 T cells? If so, do NKT express CTLA4 and PD-L1 and respond to these antibodies?

We thank the reviewer for his/her observations.  We cannot exclude that some LDH could be released also from NKT cells that could be present in both the cell subpopulations but the percentage of these cells is very low even though they express CTLA4 and PD-L1, as previously reported (see refs 47 and 48 added in the manuscript), and thus they could respond to these antibodies. We have added a paragraph in the Results section (see p. 6, lines 264-268) to clarify this point.

  1. The relevant question for comparing T cell modulation by the new antibodies versus the existing therapeutic antibodies is: how well do they increase LDH release when pan T cells obtained from breast cancer patients are incubated with their own tumor cells?  This question could be better addressed using existing CD8 T cell lines with appropriate TCRs that recognize known breast cancer antigens.

We thank the reviewer for his/her suggestions. We have used the allogenic reaction (between tumor cells and immune cells of different donors) to activate immune cells an let them express the immune checkpoints more efficiently than it may occur in isogenic reaction (between tumor cells and immune cells of the same donors). Of course it would be interesting using engineered CD8 T cell lines with appropriate TCRs that recognize breast cancer antigens, such as CAR-T cells, however they are not easy to get, they have different mechanisms of action from natural immune cells (a TCR-independent artificial immune synapse) and could not resemble the in vivo environment of cancer patients treated with conventional antibody-based therapies.

  1. Given that Fig 1A,B documents some expression of CTLA-4 and PD-L1 on the breast cancer cells, and previous publications have reported direct cytotoxic activities of these antibodies for other cell types, do the antibodies cause any LDH release when incubated with these cell lines in the absence of PBMCs?

We thank the reviewer for his/her comments. We have checked the release of LDH by treated tumor cells in the absence of immune cells and we have added two sentences relative to the results in the text (see p.8, lines 301-304 and p.10, lines 366-369).

Reviewer 3 Report

This study reported the used of two Anti-PD-L1 and Anti-CTLA-4 2 mAbs in the cell viability and cytokine secretion of lymphocytes subpopulations. This topic is interesting and the study is valuable for clinical immune therapies for cancers. However, the study requires modification to reach the quality for publication. Here are two major issues: 1. The study is all based on ELISA type assay and it is necessary to validate the main conclusion with other types of protein detection assay. 2. The quality of the figures and statistical analysis must be improved.

Specific commons:

“cancers-1957876-non-published” and “cancers-1957876-supplementary” are the same file, please check them.

Figures are not mentioned in the text.

In Cell ELISA, cells should be fixed before blocking.

In cell ELISA, usually background is very high, the negative control group must be included in the assay.

Why fig1A has two panels? Is there any specific reason for combining 231 and 549 in one?

Why the Fig1B set the control at 12 hour?

This should be explained in the text. Fig2, NK and Pan T -en-260 riched populations were treated with SEB for 66 h, the “66h” needs a reference.

The statistical analysis is not clear, if this is a turkey’s test, only part of the results were shown. This is also the same in the other figures.

Author Response

Dear Editors of Cancers,

We wish to thank you and the Reviewers for the useful comments and suggestions on our manuscript (ID: cancers-1957876) entitled "New Insights on the Role of Anti-PD-L1 and Anti-CTLA-4 mAbs on different Lymphocytes Subpopulations in TNBC ". We have fully revised the manuscript by inserting most of the requested modifications or experiments and by preparing new figures.

Please find below a point-by-point reply to the Reviewers comments (reported in italic). Major changes are highlighted by using the "Track Changes" function within the main body of the text.

This study reported the used of two Anti-PD-L1 and Anti-CTLA-4 2 mAbs in the cell viability and cytokine secretion of lymphocytes subpopulations. This topic is interesting and the study is valuable for clinical immune therapies for cancers. However, the study requires modification to reach the quality for publication. Here are two major issues: 1. The study is all based on ELISA type assay and it is necessary to validate the main conclusion with other types of protein detection assay. 2. The quality of the figures and statistical analysis must be improved.

Specific commons:

“cancers-1957876-non-published” and “cancers-1957876-supplementary” are the same file, please check them.

We thank the reviewer for his/her comments and we are sorry for the inconvenient. They are the same file indeed, we had problems when we uploaded the files in finding the window of supplementary material, thus we used the non-published one.

Figures are not mentioned in the text.

We have mentioned the figures in the text, as appropriately suggested (see p.13, lines 425-426).

In Cell ELISA, cells should be fixed before blocking.

We thank the reviewer for his/her comments. We have addedd a sentence in the Methods section to explain that the cells were used in suspension in round bottom plates, so that they are recovered by centrifugation at each step (see p.4, lines 163-165) and thus they did not need to be fixed allowing for a native conformation of the cell surface receptors that could be partially lost when a fixative such as paraformaldehyde is used.

In cell ELISA, usually background is very high, the negative control group must be included in the assay.

We thank the reviewer for his/her comments. We have included as a negative control for cell ELISA MCF7 cells, that do not express or express at very low levels the ICs, in the new Figure 1A. As for cell ELISA on lymphocytes, the control was represented by unstimulated lymphocytes. We have inserted two sentences (p. 5, lines 206-208; lines 218-223) and a new figure 1A in the Results section to clarify this point.

Why fig1A has two panels? Is there any specific reason for combining 231 and 549 in one?

We thank the reviewer for his/her comments. We have combined the panels relative to MDA-MB 231 and BT549 cells as they are TNBC, differently from BT-474 that are non-TNBC cells.

Why the Fig1B set the control at 12 hour?

We thank the reviewer for his/her comments. We have replaced the Fig.1B with the corrected one where the control is set at 0, as appropriately suggested.

This should be explained in the text. Fig2, NK and Pan T -en-260 riched populations were treated with SEB for 66 h, the “66h” needs a reference.

We thank the reviewer for his/her comments. We have inserted a sentence with a reference in the text (p.6, lines 251-253) to explain that the time of 66 hours was previously used for unfractionated lymphocytes as it was found corresponding to the peak of activation and secretion of cytokines.

The statistical analysis is not clear, if this is a turkey’s test, only part of the results were shown. This is also the same in the other figures.

We thank the reviewer for his/her comments. We have included statistics (by Student’s t-test) for single and for the combined antibodies comparisons in the new Figures 3, 4 and 5.

Round 2

Reviewer 2 Report

The revisions and added data have addressed all of the previous concerns.

Author Response

We thank the reviewer for his/her kind words on our work to address his/her previous concerns and for appreciating our paper.

Reviewer 3 Report

I am very appreciative of the response from the authors. The author has solved most of the major issues I raised. However, the author skips one of the major issues, which I think is very critical for this study to be convincing. I have mentioned that the study is all based on ELISA type assay and it is necessary to validate the main conclusion with other types of protein detection assay. This is because Cell ELISA assay is always tricky in experiments. The author’s control is not convincing. Even including negative control, it is difficult to control the cell number. It is not sure how many cells were lost during the washing. A suggested way to do it is to count cells after the ELISA. To my knowledge, cell ELISA always suffers from a strong background, ELISA is a very sensitive assay, and the color change can increase by waiting for longer. Usually, the number of ELISA results A450 can go up to 2. In this study, the number is only lower than 0.6, which can be just background from incomplete washing. This is ok if the cell ELISA is one of the multiple assays cross-validating each other’s results. Yet, this study counts only on one assay. Therefore, I suggested that this study should include other protein detection assays to confirm this conclusion. I notice that the cells were not permeabilized, this means the ELISA is testing the membrane expression. One way to validate it is to immunoprecipitate membrane protein and do a western blot. Another way to validate it is to do cell flow cytometry. I hope my concern can be addressed properly.And the paper can be accepted if this issue is solved.

Author Response

We thank the reviewer for his/her kind words on our work to address his/her previous concerns.

We understand his/her comments on ELISA as the only assay to check the level of expression of ICs, however we frequently use Cell ELISA as it allows to monitor the binding to the receptors in their native conformation and indeed is a very sensitive assay. As for the background, we use to block the cells with 5% milk or 6% BSA that strongly reduces non specific binding. Anyway, to meet the criticism of the reviewer we analyzed the expression levels of ICs on BT-474 and MCF-7 tumor cells and on lymphocytes also by Western Blotting analyses of the cell extracts, as suggested. We included these new data in the revised  manuscript in the text (see p. 5, lines 221-223 and lines 232-234) and in a new Figure (see new Supplementary Figure 1).

As for the Western blotting analyses of TNBC cells (MDA-MB231 and BT-549), they were already previously published by our group on Cancers in another paper (Vetrei C, Passariello M, Froechlich G, Rapuano Lembo R, Zambrano N, De Lorenzo C. Immunomodulatory mAbs as Tools to Investigate on Cis-Interaction of PD-1/PD-L1 on Tumor Cells and to Set Up Methods for Early Screening of Safe and Potent Combinatorial Treatments. Cancers (Basel). 2021 Jun 8;13(12):2858. doi: 10.3390/cancers13122858).

Round 3

Reviewer 3 Report

Thanks for the reply and the paper is good now.